# TLR9 Monotherapy in Immune-Competent Mice Suppresses Orthotopic Prostate Tumor Development

**DOI:** 10.3390/cells13010097

**Published:** 2024-01-02

**Authors:** Mark A. Miles, Raymond Luong, Eunice E. To, Jonathan R. Erlich, Stella Liong, Felicia Liong, Jessica M. Logan, John O’Leary, Doug A. Brooks, Stavros Selemidis

**Affiliations:** 1Centre for Respiratory Science and Health, School of Health and Biomedical Sciences, RMIT University, Bundoora, VIC 3083, Australia; 2Infection and Immunity Program, Biomedicine Discovery Institute, Department of Pharmacology, Monash University, Clayton, VIC 3800, Australia; 3Cancer Research Institute, Clinical and Health Sciences, University of South Australia, Adelaide, SA 5001, Australia; 4Discipline of Histopathology, School of Medicine, Trinity Translational Medicine Institute (TTMI), Trinity College Dublin, D8 Dublin, Ireland; 5Sir Patrick Dun’s Laboratory, Central Pathology Laboratory, St James’s Hospital, D8 Dublin, Ireland; 6Molecular Pathology Laboratory, Coombe Women and Infants’ University Hospital, D8 Dublin, Ireland

**Keywords:** prostate cancer, toll-like receptor 9, CPG-1668, immunotherapy

## Abstract

Prostate cancer is ranked second in the world for cancer-related deaths in men, highlighting the lack of effective therapies for advanced-stage disease. Toll-like receptors (TLRs) and immunity have a direct role in prostate cancer pathogenesis, but TLR9 has been reported to contribute to both the progression and inhibition of prostate tumorigenesis. To further understand this apparent disparity, we have investigated the effect of TLR9 stimulation on prostate cancer progression in an immune-competent, syngeneic orthotopic mouse model of prostate cancer. Here, we utilized the class B synthetic agonist CPG-1668 to provoke a TLR9-mediated systemic immune response and demonstrate a significant impairment of prostate tumorigenesis. Untreated tumors contained a high abundance of immune-cell infiltrates. However, pharmacological activation of TLR9 resulted in smaller tumors containing significantly fewer M1 macrophages and T cells. TLR9 stimulation of tumor cells *in vitro* had no effect on cell viability or its downstream transcriptional targets, whereas stimulation in macrophages suppressed cancer cell growth via type I IFN. This suggests that the antitumorigenic effects of CPG-1668 were predominantly mediated by an antitumor immune response. This study demonstrated that systemic TLR9 stimulation negatively regulates prostate cancer tumorigenesis and highlights TLR9 agonists as a useful therapeutic for the treatment of prostate cancer.

## 1. Introduction

Prostate cancer is the second most frequent cancer in men, with over 300,000 patients dying each year, largely due to the limited treatment strategies available for advanced-stage disease [1,2]. Excessive and unbalanced inflammation is a key contributor to prostate cancer genesis and pathogenesis, which may be exacerbated by chronic inflammation from various pathogens, hormone changes, lifestyle, and other environmental factors [3]. A first line of defense against bacterial and viral infections is the innate immune system, and the recognition of these pathogens relies on pattern-recognition receptors such as toll-like receptors (TLRs) [4]. TLRs achieve this by recognizing unique molecular patterns at either the cell surface or within endosomal compartments. TLR activation facilitates an immediate defense response via the innate immune system and subsequently is used to initiate an adaptive immune response as a clearance mechanism and to protect against subsequent infections [5]. Cell-surface TLRs are mainly responsible for the detection of bacterial pathogens or danger-associated molecular patterns (PAMPs/DAMPs), while endosomal TLRs, such as TLR3, 7, 8, and 9, are predominately activated upon the recognition of endocytosed antigens, either PAMPs or DAMPs, like viral nucleic acids [4]. In general, TLR ligation leads to a proinflammatory response mediated via the TIR domains of the receptor and the MyD88 or TRIF adaptor proteins, which initiate a cascade of downstream signals, such as NFκB, mitogen-activated protein kinases, and interferon regulatory factors, to produce their immunological effects [6].

In addition to the involvement in bacterial and viral defense, TLRs have been implicated in both the promotion and inhibition of cancer pathogenesis, and these apparently opposing roles have been rationalized upon examining various types of stimulated TLRs and cancer types [7]. Bacterial or pharmacological stimulation of TLR2, 4, and 5 have all been shown to be beneficial in the treatment of a myriad of cancers, such as bladder, uterine, and breast cancer [8,9,10,11,12]. In prostate cancer, *in vitro* activation of TLR4 by LPS promoted proliferation and survival of prostate epithelial PC3 cancer cells through the VEGF and TGF-β dependent pathways [13], while TLR4 silencing via small interfering RNA showed the opposite effect [14]. TLR3 stimulation in LnCAP and PC3 prostate cancer cell lines inhibited cell proliferation via the protein–kinase C pathway whilst simultaneously inducing caspase-dependent apoptosis [15]. Additionally, poly I:C (the pharmacological agonist to TLR3) has been shown to directly induce apoptosis in breast cancer cells [16]. These diverse findings indicate a need to further understand the biology of TLRs, which may then identify potential strategies for therapeutic intervention in cancer patients.

In this study, we investigated the role of TLR9, which is responsible for the detection of unmethylated cytosine–guanine dinucleotide (CPG) sequences in DNA from bacteria and viruses [17,18] within the endosomal compartment. TLR9 is primarily expressed in antigen-presenting cells, although its expression has also been found in some nonimmune and cancer cells. Like TLR4, there is division surrounding the role of TLR9 activation in cancer pathogenesis. Increased expression of tumor-specific TLR9 has been associated with higher tumor grade and poor prognosis in patients suffering from breast, ovarian, or prostate cancer [19,20]. In prostate cancer, overexpression or pharmacological stimulation of TLR9 by synthetic CPG oligodeoxynucleotides (CPG-ODNs) has been shown to induce invasion through the upregulation of proangiogenic and protumorigenic signals, such as MMP9, MMP13, IL-8, and TGF-β, and this can allow for immune evasion by the tumor [21,22,23,24,25]. Prostate cancer cells that express low or no TLR9 fail to undergo an invasive phenotype upon CPG-ODN treatment, emphasizing TLR9-specific signaling within the tumor cell as an important driver for enhancing cancer progression. Indeed, silencing of TLR9 by siRNA reduced subcutaneous hematological tumors by half [26]. Additionally, a subcutaneous prostate cancer model utilizing NOD SCID gamma mice showed that tumor growth correlated with the expression of TLR9 in tumor cells [27]. This effect was dependent on downstream TLR9 signaling, leading to NFκB and STAT3 activation, but this involved an immune-compromised animal model. Hence, the capacity for TLR9 stimulation to modify immune activation would be underestimated. Indeed, TLR9 agonists can enhance antitumor immunity by reprogramming the tumor microenvironment and increasing the tumor-suppressive potential and/or cytotoxic activity of innate and adaptive immune cells [28,29], leading to immune rejection of solid tumors [30,31,32,33]. These antitumorigenic effects are often associated with increased production of tumor necrosis factor alpha (TNFα) and type-1 interferons (IFN) from TLR9-stimulated cells [34,35]. This suggests that TLR9 activation in immune cells could also be favorable in prostate cancer, although the evidence for this remains experimentally unclear probably due to a lack of prostate cancer models that employ orthotopic tumor implantation. Overall, the apparent dual function of TLR9 to act in immune cells (to promote antitumor immunity) or in cancer cells (to promote tumor growth and progression) will likely govern the outcomes of therapeutic TLR9 stimulation in various cancer models, including prostate cancer. As such, the combined inhibition of STAT3, a protumorigenic signal activated in TLR9-stimulated tumor cells, with CPG-ODNs led to the suppression of subcutaneous or intratibial castrate-resistant prostate tumors in mice [36]. This model offers a therapeutic strategy to suppress the STAT3-mediated tumor survival effect in prostate cancer cells that is associated with TLR9 stimulation alongside boosting the antitumor immunity to skew the TLR9 activation response in favor of tumor killing.

Another reported effect of TLR9 agonism in prostate cancer has been the direct TLR9-dependent killing of tumor cells in some settings. RM1 murine prostate cancer cells treated with CPG-ODNs underwent cell death via a mechanism that blocked the prosurvival effects of NFκB and AP-1 [37]. This can additionally enhance the antitumor responses by providing an internal source of tumor antigen to improve immune cytotoxicity. Indeed, CPG-ODN in combination with tumor lysate delivered via microspheres significantly blunted prostate cancer development in a transgenic adenocarcinoma mouse prostate (TRAMP) model [38].

To address these discrepancies, we utilized a syngeneic model of prostate cancer to examine the impact of TLR9 stimulation on the growth of orthotopic prostate tumors. A limitation of many prior studies is that they employ a subcutaneous or immunocompromised prostate tumor model, which may not necessarily reflect complex interactions that occur within the inflammatory tumor microenvironment of the prostate. Furthermore, immunocompromised models may underestimate the global immunostimulatory effects of TLR9 activation. We tested the synthetic TLR9 agonist class B CPG-ODN 1668 (CPG-1668), which contains 1-5 CPG motifs on a phosophorothionate backbone to enable a half life of 30–60 min [39,40]. Taking this short half life into consideration, we delivered CPG-1668 via an osmotic minipump to mice bearing RM1 prostate tumors to allow for constant infusion daily over a 14-day period. We observed that systemic CPG-1668 administration suppressed orthotopic prostate cancers, and this was associated with enhanced systemic immunity, supporting the utility of TLR9 agonists as a therapeutic strategy for prostate cancer.

## 2. Materials and Methods

### 2.1. Orthotopic Murine Model of Prostate Cancer

Male C57Bl6/J mice between 8–12 weeks in age were obtained from Monash Animal Services (Monash University, Melbourne, Australia) and housed in an approved holding facility with ad libitum access to water and standard rodent chow (4.8% fat, 0.02% cholesterol). All experimental procedures were approved by the Monash University Animal Ethics Committee.

The mice were anesthetized via inhalation of a 1–5% isoflurane and 95% oxygen air mixture and subcutaneously injected with Carprofen (5 mg/kg). Once reflex to toe pinch was absent, a small incision (~1 cm) was made through the skin and muscle of the abdomen to access the prostate. A cell suspension (10 μL) containing 5 × 10^3^ RM1 cells in DMEM (Invitrogen, Carlsbad, CA, USA) plus 10% Foetal Bovine Serum (FBS; Sigma, Bayswater, VIC, Australia) was injected into the ventral prostate. Sham mice received 10 μL of DMEM plus 10% FBS only. Some RM1 cells were pretreated with PBS or 10 µg/mL CPG-1668 (InvivoGen, San Diego, CA, USA) for 1 h prior to engraftment. An osmotic minipump containing PBS or CPG-1668 (to enable the delivery of 50 µg/day) was also implanted portal first along the left scapular line of the mouse immediately after tumor engraftment surgery. The tumors were allowed to develop for 14 days before mice were terminally anesthetized via a ketamine (180 mg/kg)/xylazine (32 mg/kg) intraperitoneal (i.p.) injection.

Prostates and spleens were dissected out for gross morphological analysis. The heart was pumped with 0.1 mL of clexane (400 U/mL) prior to blood collection via cardiac puncture. The blood and organs were then processed for flow cytometric analysis. Tumor weights were recorded as prostate and its associated tumor (or prostate only in the case of sham mice) plus seminal vesicles.

### 2.2. Flow Cytometric Analysis of Immune Populations

Prostate samples were minced using scissors and enzymatically digested in a digestion buffer consisting of collagenase type XI (Sigma), hyaluronidase (Sigma), and collagenase type I-S (Sigma) dissolved in PBS (containing Ca^2+^ and Mg^2+^) for 45 min at 37 °C shaking. Spleens were minced thoroughly using scissors without enzymatic digestion. Total blood leukocytes were isolated from the whole clexane-mixed blood. Osmotic lysis of excess red blood cells (RBC) for all suspensions was performed using RBC buffer (NH_4_Cl 155 mM, NaHCO_3_ 12 mM, and EDTA 0.1 mM), washed using PBS and with the cells pelleted. Samples were then passed through a 70 µm sterile cell strainer and washed twice with fluorescence-activated cell-sorting (FACS) buffer (PBS supplemented with 5% FBS) to yield a single cell suspension. Cells were counted, resuspended in PBS containing aqua live/dead viability stain (Life Technologies, Carlsbad, CA, USA), and incubated for 15 min at 4 °C. Cells were then washed and stained with fluorophore-conjugated antibodies specific for surface markers using the following antibody panel from Biolegend (San Diego, CA, USA) unless otherwise stated: PerCP anti-CD45 (30-F11, cat #103130); APC anti-CD3 (145-2C11, BD Biosciences, Macquarie Park, NSW, Australia, cat #553066); PE anti-CD8 (53-6.7, BD Biosciences cat #553032); BV605 anti-CD4 (RM4-5, BD Biosciences cat #100548); APC-Cy7 anti-Ly6G (1A8, cat #127624); FITC anti-CD69 (H1.2F3, BD Biosciences cat #557392); APC anti-iNOS (CXNFT, Invitrogen cat #53-5920-80); PE-Cy7 anti-CD206 (C068C2, cat #141719); and PE anti-F4/80 (BM8, Thermo Fisher Scientific, Waltham, MA, USA, cat #12-4801-82). Following immunostaining, cells were resuspended in FACS buffer, fixed, and analyzed the following day on a FACS ARIA flow cytometer with the FACS DIVA software v6.1.3(BD Bioscience). Countbright counting beads (Life Technologies), were added to each sample and a minimum of 10,000 beads were collected to determine the cell number. Data were analyzed using FlowJo software (Tree Star, Inc., Ashland, OR, USA). The representative gating strategy is shown in Appendix A.

For TLR9 staining, cells were incubated with mouse anti-TLR9 antibody (26C593.2, Abcam (Waltham, MA, USA), cat #ab134368) for 1 h at 4 °C, followed by 1 h incubation with donkey antimouse IgG-AlexaFluor488 (Abcam cat #ab150105). Prior to staining, cells were either fixed using the Foxp3 Transcription Factor Staining Buffer Set (Thermo Fisher Scientific) to measure intracellular TLR9 or not fixed to measure the surface TLR9 protein.

### 2.3. Cell Culture

RM1 cells, an immortalized murine prostate carcinoma androgen-insensitive cell line derived from the genital ridge of C57BL6/J mice, were grown in DMEM (Invitrogen) supplemented with glucose (1 g/L), 2 mM L-glutamine, phenol red, and 10% FBS. Immortalized bone-marrow-derived macrophages (BMDMs) were provided by Prof. Ashley Mansell (Hudson Institute of Medical Research, Monash University, Australia) and Prof. Eicke Latz (Institute of Innate Immunity, University of Bonn, Germany) and grown in DMEM supplemented with glucose (4.5 g/L), 2 mM L-glutamine, phenol red, and 10% FBS. The immortalized normal human prostate epithelial PNT1A cells were provided by Prof. Roger Daly (Department of Biochemistry and Molecular Biology, Monash University, Australia) and were maintained in RPMI-1640 (Invitrogen) media supplemented with 2 mM L-glutamine, phenol red, and 5% FBS. Human prostate adenocarcinoma LNCaP cells were provided by Prof. Tony Tiganis (Department of Biochemistry and Molecular Biology, Monash University, Australia) and were maintained in RPMI-1640 media supplemented with 2 mM L-glutamine, 10 μM HEPES, 10 μg human insulin, phenol red, 1 mM sodium pyruvate, and 10% FBS.

### 2.4. Cell Proliferation

To determine clonogenic potential, 1 × 10^4^ RM1 cells were seeded in 24-well plates and left to adhere overnight. Cells were then treated with recombinant murine IFN-β (In Vitro Technologies, Noble Park, VIC, Australia), IFN-γ (Invitrogen), TNFα (Peprotech, Cranbury, NJ, USA), CPG-1668, or BMDM-conditioned media. After 24 h incubation, cells were harvested and counted, after which, 100–300 cells were seeded per well in a 6-well plate. The plates were stained after 7–8 days with 1% crystal violet (Sigma), and the number of colonies was counted. In some experiments, cells were pretreated with 50 µg/mL antimouse IFNAR1 blocking monoclonal antibody (BioXcell, Lebanon, NH, USA) prior to treatment. BMDM-derived conditioned media was generated by culturing 1.5 × 10^5^ cells per 12-well in CPG-1668 or not for 1 h, washing the compound off with PBS, and then replacing it with fresh media. After incubating cells for a further 23 h, conditioned media was collected, and any cell debris was pelleted before storing at −80 °C until use.

To assess cell proliferation, 2.5 × 10^4^ RM1 murine prostate cancer cells, 1 × 10^5^ LnCAP human prostate cancer cells, or 5 × 10^4^ PNT1A normal human prostate cells were seeded into 6-well cell-culture plates and left to adhere overnight. Cells were treated with PBS, CPG-1668 (10 µg/mL), or IFN-β daily for 24, 48, and 72 h. Media and drugs were replenished daily. Cells were washed with PBS and lifted using trypsin for quantification via trypan blue exclusion (Thermo Fisher Scientific).

### 2.5. Cell Survival

Cell viability was measured using the CellTiter 96 Aqueous One Solution Cell Proliferation Assay (MTS) kit (Promega, Alexandria, NSW, Australia). Briefly, 1 × 10^3^ cells were seeded in 96-well plates a day before treatment; then, the media was replaced with 100 µL fresh media containing cytokines or not. After 24-, 48-, or 72-h of incubation, 20 µL MTS reagent were added per well, and the plates were incubated at 37 °C for 1 h before measuring absorbance at 490 nm using a ClarioStar Plus Microplate Reader (BMG Labtech, Mornington, VIC, Australia).

To determine the degree of cell death, 1 × 10^4^ RM1 cells were seeded in 24-well plates and then treated the next day with CPG-1668 or IFN-β. The media and cells were harvested, then resuspended in PBS (containing Ca^2+^ and Mg^2+^)-containing AnnexinV-FITC (1:100; Biolegend) and incubated at room temperature for 15 min. Propidium iodide (Biolegend) diluted in PBS was then added to a final concentration of 1 µg/mL and the cells were immediately analyzed on a BD LSRFortessa X-20 flow cytometer with DIVA software v6.1.3 (BD Biosciences). Data analysis was performed using FlowJo software (Tree Star, Inc.).

### 2.6. Gene Expression by Real-Time qPCR

RM1 (1 × 10^5^) or BMDM (2 × 10^5^) cells were seeded in 12-well plates overnight and then treated with increasing concentrations of CPG-1668 for 24 h. Total RNA was extracted using an RNeasy Mini Kit (Qiagen, Hilden, Germany). Then, cDNA was synthesized using the High-Capacity cDNA RT kit (Thermo Fisher Scientific) from 2 µg total RNA according to the following protocol: 25 °C for 10 min, 37 °C for 120 min, 85 °C for 5 min, and kept at 4 °C until collection using the Veriti Thermal Cycler (Applied Biosystems, San Francisco, CA, USA). Quantitative polymerase chain reaction (qPCR) was carried out using the TaqMan Fast Advanced Master Mix (Thermo Fisher Scientific) and analyzed on an Applied Biosystem QuantStudio 7 Flex Real-Time PCR System (Thermo Fisher Scientific). Assay On-Demand Gene Expression Assay Mix primers for *IL-6*, *IL-1β*, *IFN-β*, *TNF-α*, *IFN-γ*, *MMP9*, *TGF-β*, *VEGF*, *NFκB*, *STAT3*, and *GAPDH* were used. Quantitative values were obtained from the threshold-cycle (Ct) number and gene-expression analysis was performed using the comparative Ct method. The target gene-expression level was normalized against *GAPDH* mRNA expression for each sample and data was expressed relative to appropriate controls.

### 2.7. Protein Quantification

Cells were lysed in RIPA buffer (150 mM sodium chloride, 1.0% Triton X-100, 0.5% sodium deoxycholate, 0.1% SDS, 50 mM Tris, pH 8.0) containing a protease-inhibitor cocktail (Sigma) and supernatants cleared by centrifugation at 10,000× *g* for 15 min at 4 °C. Protein levels in lysates were quantitated using the Pierce BCA Protein Assay Kit (Thermo Scientific) according to the manufacturer’s instructions. Thirty micrograms of protein were then separated by SDS-PAGE, transferred to PVDF membranes (BioRad Laboratories, Hercules, CA, USA), and then membranes blocked in 5% skim milk for 1 h at room temperature. Mouse anti-TLR9 (Abcam, Cambridge, UK, cat #AB134368) or rabbit anti-GAPDH (Cell Signaling Technology, Danvers, MA, USA, cat #5174) antibodies diluted in TBS-T were added and the membranes incubated overnight at 4 °C, washed, and then probed with HRP conjugated secondary antibodies (Cell Signaling Technology). Chemiluminescent detection was achieved with the SuperSignal West Femto Maximum Sensitivity Substrate (Thermo Fisher). ImageJ was used for the densitometry analysis of bands.

IFN-β levels in cell-culture media were measured using the Mouse IFN-beta DuoSet ELISA kit (R&D Systems, Minneapolis, MN, USA) according to the manufacturer’s instructions.

### 2.8. Statistical Analysis

GraphPad Prism 8.0 ( Boston, MA, USA) was used to perform unpaired, two-sided *t*-tests, or one- or two-way ANOVA analyses with post hoc tests for multiple comparisons (outlined in the figure legends). All values of *p* < 0.05 were considered to indicate statistical significance (Appendix A). The results are expressed as mean ± standard error of the mean (SEM).

## 3. Results

### 3.1. Systemic CPG-1668 Administration Murine Prostate Cancer Tumorigenesis

Utilizing an orthotopic syngeneic *in vivo* model of prostate cancer [41], C57BL6/J mice were injected with either the vehicle or RM1 cells into the ventral lobe of the mouse prostate. RM1-engrafted mice developed sizeable tumors in their prostates resulting in significantly heavier prostates (~70%) compared to the sham controls (Figure 1A,D). On the same day of tumor engraftment surgery, all mice were subcutaneously implanted with osmotic minipumps along the left scapular line of each mouse, and these contained either PBS (as a vehicle) or CPG-1668 (enabling the delivery of 50 µg per day). An additional group was included whereby RM1 cells were pretreated for 1 h with CPG-1668 (10 µg/mL) prior to engraftment surgery, and these mice were implanted with minipumps containing PBS only. Constant infusion of CPG-1668 resulted in significantly suppressed tumor development compared to mice that received a constant infusion of PBS, which was consistent with the difference in prostate weights (Figure 1A,D). Mice engrafted with CPG-1668 pretreated RM1 cells developed sizeable tumors comparable to RM1 control mice, which was evident by the increased prostate weight compared to the sham control. This implied that constant infusion of CPG-1668 *in vivo* was required to exert an antitumor effect and that this was likely due to an enhanced immune response. Indeed, mice that received constant TLR9 stimulation developed splenomegaly (Figure 1B) indicating a systemic immune response, although there was no change in overall bodyweight across all treatment groups (Figure 1C).

These data demonstrated that daily stimulation of TLR9 suppressed the development of RM1 prostate tumors, but pretreating RM1 cells with CPG-1668 prior to engraftment had no effect. This suggested that the *in vivo* TLR9 response probably invoked a tumor-suppressive immune response, rather than directly causing the cell death of RM1 cells. We, therefore, decided to inquire further into the role of TLR9 activation and its effect on immune cells, which could be causing the reduction of prostate tumor size.

### 3.2. CPG-1668 Treatment Reduces T Cell and M1 Macrophage Populations in Prostate Tumors

Flow cytometry was used to evaluate the immune cells present at the experimental endpoint in the prostate tissue harvested from RM1 tumor-bearing mice continuously treated or not treated with CPG-1668. We observed approximately 10^6^ white blood cells in untreated prostate tumors consisting of T cells, macrophages, and neutrophils (at least from the antibody panel used for this study). CPG-1668 treatment significantly reduced the frequency and number of prostate-specific leukocytes by more than half (Figure 2A) indicating that, while these tumors were smaller in size, they also contained fewer immune cells. Strikingly, prostates from CPG-1668-treated mice contained severely low numbers of T cells, including the CD4+ and CD8+ subsets compared to untreated tumors (Figure 2B). CD8+–CD4+ T-cell ratios revealed that untreated prostate tumors contained twice as many CD8+ T cells than CD4+ T cells, while CPG-1668 treatment resulted in more CD4+ T cells, although the cell numbers here were very low. Additionally, most of the T cells detected in untreated tumors stained positive for CD69, suggesting they were “activated” despite the substantial tumor size. It is possible that the T cells infiltrating prostate tumors contribute to tumor maintenance rather than its suppression, and that systemic TLR9 stimulation alters this effect. The majority of macrophages detected in untreated prostate tumors were M1 polarized, and CPG-1668 treatment significantly reduced the absolute numbers of these “inflammatory” M1 macrophages (Figure 2C). The numbers of “alternative” M2 macrophages and Ly6G+ neutrophils (Figure 2D) in the prostate were unaltered with CPG-1668 treatment, although, based on composition, these treated tumors contained more M2 macrophages and neutrophils.

### 3.3. CPG-1668 Treatment Boosts Systemic Macrophage, Neutrophil, and Cytotoxic CD8+ T-Cell Numbers

An analysis of the whole blood revealed no differences in total leukocytes with treatment (Figure 3A). Overall, CD3+ T-cell populations were also similar between mice, although there were fewer total and activated CD4+ T cells in CPG-1668-treated mice compared to PBS control mice (Figure 3B). Interestingly, CPG-1668 treatment boosted the frequency of total and activated CD8+ T cells. We observed a significant increase in the ratio of CD8+ to CD4+ T cells in mice that received CPG-1668 compared to untreated mice (Figure 3B), indicating that more cytotoxic T cells were in circulation. This observation contrasts with the T-cell ratios in the prostate and may reflect the persistent immunostimulation induced by continual CPG-1668 delivery. Indeed, CPG-1668 treatment also enhanced the number of circulating F4/80+ macrophages, of either M1 or M2 subtypes (Figure 3C), and neutrophils (Figure 3D). According to M1–M2 ratios, the majority of circulating macrophages in tumor-bearing mice were M1 like, and this appeared to increase slightly with CPG-1668 treatment.

Given the spleen plays a critical role in innate and adaptive immune responses, we also performed flow cytometric analysis on the spleens from these mice. As expected, CPG-1668 treatment enhanced the total number of leukocytes in the spleen (Figure 4A), consistent with splenomegaly observed in these mice. The spleens of CPG-1668-treated mice contained lower frequencies of CD4+ and CD8+ T cells, although no significant alterations in the number of splenic T-cell populations were detected (Figure 4B). The difference is probably a reflection of the larger spleen sizes in these mice. Interestingly, there were significantly more frequent CD69-stained CD8+ T cells following CPG-1668 treatment similar to the systemic observations. Treatment also resulted in a small but significant increase in the CD8+–CD4+ T-cell ratio. Consistent with the systemic analysis, splenic macrophage (Figure 4C) and neutrophil (Figure 4D) numbers were dramatically boosted with CPG-1668 treatment, suggesting enhanced systemic innate immunity as well as cytotoxic CD8+ T cells in these mice.

These data reveal that TLR9-mediated suppression of prostate tumor growth largely alters T-cell and M1 macrophage populations within the prostate. We also found enhanced gene expression of the inflammatory cytokines *IFN-β*, *IFN-γ*, *IL-18*, and *IL-10* in the spleens of naïve mice treated with CPG-1668 (Appendix A), suggesting that TLR9 agonism induces systemic inflammatory signaling. Therefore, the enhanced systemic innate and CD8+ T-cell immune activation, combined with the smaller tumor weights containing fewer immune cells, implies that TLR9 stimulation suppressed prostate immune infiltration or possibly altered the state of tumor-residing immune cells towards an antitumor phenotype.

### 3.4. In Vitro Stimulation of TLR9 in Immune Cells but Not RM1 Cells Limit Cancer Cell Proliferation

The data thus far suggest that the TLR9-dependent reduction in prostate tumorigenesis is likely the result of altered immune activation and subsequent antitumor immunity, rather than direct effects on the viability of RM1 cells grafted into mouse prostates (as immune-cell populations were enhanced systemically, and CPG-1668 pretreatment in RM1 cells failed to have any impact). We next compared the transcriptional responses to direct CPG-1668 exposure in RM1 cells or macrophages, which are known to express TLR9. Significant upregulation of the proinflammatory genes *IL-6*, *IL-1β*, and *IFN-β* was detected upon TLR9 stimulation of bone-marrow-derived macrophages (BMDMs) although the expression of these genes remained unchanged in RM1 cells, except for *IL-1β* at the highest CPG-1668 dose tested (Figure 5A). *In vitro* stimulation of TLR9 in prostate cancer lines enhances the expression of protumorigenic and pro-proliferative genes [21,23,27]; however, the response of RM1 cells to CPG-1668 specifically has not been reported. We did not observe any significant changes in *MMP9*, *TGF-β*, *VEGF*, *NFκB*, or *STAT3* gene expression in RM1 cells or BMDMs following treatment with CPG-1668, although *NFκB* was downregulated in BMDMs (Figure 5B). Additionally, analysis of RM1 cells at earlier time points revealed no difference in gene expression, indicating CPG-1668 had minimal effect regardless of dose or exposure time (Appendix A). There were no significant differences in the total (Figure 5C), surface- or intracellular-specific (Figure 5D) TLR9 protein expression between RM1 cells and BMDMs. This implies that the limited response to TLR9 agonism with CPG-1668 in RM1 cells was not due to low TLR9 expression.

To address whether TLR9 stimulation by CPG-1668 directly impacts cell viability, RM1 cells were treated *in vitro*, and cell death was determined by annexin-V/propidium iodide staining or colony-forming assays. RM1 cells remained viable after 48 h incubation and maintained similar colony-forming capacity regardless of TLR9 stimulation (Figure 6A,B). The effect of CPG-1668 on cell proliferation was also assessed in RM1 cells (mouse prostate cancer), as well as LnCAP human prostate cancer cells, and PNT1a nonmalignant normal human prostate cells. All PBS-treated cultures increased in cell numbers over the course of 72 h. CPG-1668 treatment (10 µg/mL/day) over this period did not alter proliferation in any of the cell lines (Figure 6C). This finding supports the lack of effect that CPG-1668 pretreatment had on RM1 cells prior to *in vivo* engraftment; it neither caused tumor cell death nor accelerated tumor growth. This further implicates enhanced antitumor immune stimulation upon CPG-1668 delivery *in vivo* as the mechanism of tumor suppression rather than a direct effect on RM1 cells.

Upon stimulation, immune cells can secrete effector cytokines, such as TNFα or IFN-γ, to directly provoke the cell death of cancer cells [42]. Given that we observed an increase in mRNA gene expression of *IFN-β*, and to a lesser extent *TNFα* and *IFN-γ*, in BMDMs following CPG-1668 treatment, we assessed whether exposure to these recombinant cytokines had any direct impact on RM1 cell survival. Treatment of RM1 cells with IFN-β but not TNFα or IFN-γ reduced RM1 cell viability (based on MTS absorbance) up to 72 h (Figure 7A). The effect of IFN-β was not due to cell death (Figure 7B) but the result of a dose-dependent reduction in proliferation and this impacted the ability of RM1 cells to form colonies (Figure 7C,D). Exposure to TNFα or IFN-γ did not significantly alter the clonogenicity of RM1 cells (Figure 7D). CPG-1668 cotreatment with IFN-β did not inhibit the reduction in clonogenicity by IFN-β, although it was mildly boosted (Figure 7E and Appendix A), indicating CPG-1668 does not alter the growth responses to cytokine stimulation in RM1 cells. We next assessed whether IFN-β secreted by stimulated immune cells could inhibit RM1 cell growth. Up to 100 pg/mL of IFN-β were detected in conditioned media (CM) from CPG-1668-stimulated BMDMs compared to nonstimulated cells (Figure 7F), confirming cytokine secretion by these cells via a TLR9-dependent mechanism (Appendix A). Strikingly, RM1 cells grown in this CM exhibited reduced colony-forming potential (Figure 7G). The observed magnitude of RM1 clonogenic loss when grown in TLR9-stimulated CM was equivalent to treatment with 0.1–1 ng/mL recombinant IFN-β. To elucidate if the IFN-β secreted from macrophages was in fact responsible for reducing RM1 cell clonogenicity in this context, RM1 cells were incubated with an IFNAR1 monoclonal blocking antibody and then grown in CM. The blocking of IFNAR1 in RM1 cells prevented the reduction in clonogenicity or proliferation caused by growth in IFN-β or TLR9-stimulated CM (Figure 7H).

Overall, these data further support the notion that CPG-1668-mediated TLR9 stimulation in immune cells limits RM1 tumor growth via the secretion of type I IFN, leading to reduced RM1 cell proliferation and that RM1 cells fail to directly respond to CPG-1668.

## 4. Discussion

TLRs are either positive or negative regulators of cancer pathogenesis, and this may depend on the specific receptor activated, as well as tumor type and stage [7]. In this study, we provided evidence that TLR9 activation via CPG-1668 treatment markedly suppressed prostate tumorigenesis in an orthotopic *in vivo* mouse model of prostate cancer. Constant infusion of CPG-1668 resulted in significant reductions in tumor weight and boosted systemic innate immunity. Indeed, TLR9 activation via CPG-ODNs could provide protection from cancer by activating dendritic cells to stimulate innate immune responses, such as enhanced natural killer-mediated tumor killing, and the subsequent adaptive T cell and humoral immunity [43,44]. We aimed to address two possible mechanisms to explain our observations: the stimulation of TLR9 in immune cells to promote antitumor immunity capable of suppressing tumor development, or the TLR9-dependent cell death or direct growth inhibition of RM1 cells.

T cells, macrophages, and neutrophils are among the key immune-cell types that are known to infiltrate the tumor microenvironment [45,46,47]. The tumor microenvironment is highly reactive and plays a critical role in both the genesis and maintenance of prostate cancer [48,49]; thus, changes in this delicate balance could either promote or inhibit prostate tumorigenesis. Indeed, we observed a marked increase in immune cells post-tumor engraftment within the prostate, which was significantly reduced upon CPG-1668 treatment and correlated with smaller tumor weights. Prostate tumors often harbor a “cold” immunosuppressive tumor microenvironment and immunotherapies to overcome the multiple mechanisms of immuno-resistance are being trialed [50]. Recent insights into the immune profile of prostate tumors reveal a microenvironment rich in immunosuppressive myeloid cells as well as T-cell exhaustion signatures, such as increased PD-1 or PD-L1 [51,52]. As such, anti-PD-1 immunotherapy was able to improve T-cell effector activity and reduce prostate tumor development, further highlighting the immunosuppressive ability of tumor-infiltrating lymphocytes in the pathogenesis of prostate tumors [53,54]. We noted a large proportion of T cells in untreated tumors staining positive for CD69. While this is a classic marker for early leukocyte activation, CD69 expression can allow the retention of T cells within inflamed tissues to exacerbate inflammation and tumor progression [55,56], and this observation suggests that the T cells present within the tumor were restricted in their effector function.

Our data demonstrate that the TLR9-mediated inhibition of tumorigenesis in our model is likely due to shifting the tumor microenvironment in favor of antitumor immunity in the early stages of tumor development. This could be achieved through various TLR9-dependent mechanisms, for instance by encouraging the differentiation of tumor-specific myeloid-derived suppressor cells to overcome their immunosuppressive effects and boost effector CD8+ T-cell activity [57], or by directly increasing cytokine production on tumor antigen-activated CD8+ T cells [27,58,59,60]. TLR9 agonism may, therefore, bypass the need for alternative immunotherapies, such as checkpoint inhibitors, in some contexts to enhance both innate and adaptive immune responses. Moreover, TLR signaling can promote M1 polarization, which can mediate tumor destruction via the secretion of proinflammatory factors, such as NFκB, IFN-γ, and TNFα, that directly inhibit cancer cell growth [61]. We administered CPG-1668 at the same time when engrafting RM1 cells into the prostate; so, while it is possible for treatment to promote a more immunoreactive tumor microenvironment, it might also be the case that early systemic immune activation was strong enough to suppress the initial growth and prevent the complete establishment of the tumor. Healthy prostate tissue is rich in mononuclear myeloid cells and T cells [62], so TLR9 stimulation in these resident cells could promote cytokine responses to create a tumor-suppressive microenvironment and inhibit naïve tumor growth. Indeed, we found that CPG-1668 stimulated type I IFN release from macrophages, and RM1 cells grown in macrophage-conditioned media had reduced proliferative capacity. This effect was dependent on IFN-β and suggests that TLR9-activated macrophages in the vicinity of engrafted tumors could suppress tumor growth via this mechanism. As a result, the chemotactic signals that would normally recruit immune cells to the site of tumor engraftment would be absent, hence the observed lower number of immune-cell infiltrates in the smaller prostates of treated mice. We found almost no T cells in the prostates of CPG-1668 treated mice at the experimental endpoint, and this may reflect a lack of initial infiltration to help establish the tumor or may be the result of an early and rapid influx of antitumor T-cell populations to the tumor, which eventually diminished overtime to the observed levels in the smaller tumor. This might also explain why treated mouse prostates contained significantly fewer M1 macrophages.

Given that we detected significantly more circulating CD8+ than CD4+ T cells, it is likely that systemic TLR9 stimulation boosts global immunity, which consequently may act on the tumor. Similarly, CPG-1668 treatment increased both M1 and M2 macrophage numbers systemically, although it did not alter the M1-dominant phenotype (based on M1–M2 ratios) in the blood. Tumors were also M1-skewed regardless of treatment, so perhaps TLR9 stimulation helped to elevate total circulating M1 macrophage numbers to infiltrate and help redirect tumor-residing M1 macrophages towards a tumor-suppressive phenotype [63,64]. Systemic and splenic neutrophils also increased with CPG-1668 treatment, which is consistent with enhanced TLR9-mediated neutrophil viability and recruitment [65,66,67], although the numbers in tumors remained unchanged. Neutrophils are classically thought to promote the progression of cancer pathogenesis via proangiogenic secretions and excessive ROS production [68,69]. However, neutrophils are also capable of tumor clearance via the generation of cytotoxic compounds, such as H_2_O_2_ or nitric oxide [47]. Based on the consistent numbers of neutrophils in the prostate, it is unclear whether these systemic increases contributed effectively to the tumor clearance observed in our study. Prostate cancer is notorious for its plasticity allowing it to develop mechanisms for its self preservation, so it is possible that the proangiogenic properties of neutrophils in cancer [68] may be a tumor-intrinsic compensatory mechanism in response to the tumor-suppressive effects of other TLR9-activated immune cells. Further studies are needed to examine the temporal changes in prostate tumor composition driven by TLR9 stimulation in this model, and how “stimulated” immune cells function during early tumor development.

TLR9 signaling recruits the adaptor MyD88, which induces the production of type I IFNs and TNFα that are capable of inhibiting tumorigenesis [34]. Our analysis identified IFN-β as a possible driver of tumor suppression in our *in vivo* model. Type I IFN-mediated tumor suppression, via direct tumor cell inhibition or indirectly by antitumor immune responses, has been previously reported in various cancers, although the clinical efficacy of administering recombinant IFNs as therapeutics is limited by systemic toxicities and tolerability [70,71,72]. Type I IFN was recently reported to be crucial for the immune surveillance of metastatic prostate cancer, and the reactivation of type I IFN signaling reduced outgrowth of metastasis in bone [73]. Furthermore, the delivery of mesenchymal stem cells constitutively expressing IFN-β reduced the growth of metastatic prostate cancer in the lungs and increased antitumor immune activity [71]. Our study, therefore, provides a possible alternative avenue to boost antitumor IFNs via immune-cell secretion upon TLR9 stimulation, further supporting an antitumor role that TLR9 exhibits in establishing a tumor-suppressive microenvironment.

We also considered the possibility that CPG–TLR9 interactions could inhibit tumor growth by directly provoking tumor cell death. Multiple studies report that high expression of TLR9 or its stimulation with CPG-ODNs in prostate cancer cells increases the expression of invasive and metastatic genes [21,22,23,27,74]. The impact of CPG-1668 treatment on RM1 murine cells, however, has not yet been reported. We found no change in the expression of protumorigenic genes nor changes in cell growth or viability in RM1 cells directly treated with CPG-1668. Furthermore, CPG-1668 treatment of RM1 cells presurgery had no effect on tumor development. RM1 cells expressed TLR9 protein to levels comparable to macrophages, which themselves were highly responsive to TLR9 stimulation. It is possible for the RM1 cells used in our study to exhibit limited TLR9 signaling, as we only observed elevated *IL-1β* transcripts in cells exposed to high concentrations of the agonist. CPG-1668 had no effect on the growth of human prostate cancer (LnCAP) and nonmalignant cell lines, suggesting this observation was not specific to the RM1 cell line, although it should be noted that CPG-1668 has a higher specificity for mouse TLR9 [75]. We were further intrigued to observe a lack of response to CPG-1668 in RM1 cells, given that apoptosis was previously reported in these cells upon direct TLR9 activation [37]. Various CPG-ODN compounds differ in their chemical composition and TLR9–CPG retention times [40]. The CPG sequences that Shen, Waldschmidt, Zhao, Ratliff, and Krieg [37] found to provoke cell death in RM1 cells were class A agonists such as CPG-ODN 1585. These are apparently retained longer in early endosomes, allowing for better MyD88–IRF-7 interactions and signaling, and may allow for cell death pathways to be engaged. We used CPG-1668, which is a class B ODN that is rapidly trafficked to late endosomes [76]. The TLR9 signaling cascade is apparently weaker in late endosomes [76], so it could be that the endosomal turnover in RM1 cells limits sufficient TLR9 activation following stimulation with CPG-1668 to have a pronounced transcriptional effect. Even though we employed constant infusion of CPG-1668 over the course of the experiment, this method probably had a more dramatic TLR9 stimulatory effect in immune cells rather than in tumor cells, which was probably important for augmenting the host-mediated antitumor immunity that resulted in tumor suppression. It will be informative for future studies to test various CPG-ODN compounds also in orthotopic prostate tumor models to fully evaluate the effects of TLR9 stimulation on prostate cancer pathogenesis.

In summary, we have identified that stimulating TLR9 via the class B ODN CPG-1668 may be a possible therapeutic option for prostate cancer. While the immunostimulatory capacity of CPG-ODNs to enhance antitumor immunity has led to its clinical evaluation often in combination with other anticancer agents, outcomes from early phase clinical trials as a monotherapy against various cancers have been modest resulting in a CPG-ODN yet-to-be approved for therapeutic use [77]. Many of these trials used subcutaneous delivery, so the immunostimulatory effect may be underestimated, considering the short half-life of these synthetic sequences. Studies using intravenous administration have not yielded better responses, although lymphocyte activation was reportedly greater compared to subcutaneous delivery [78]. Therefore, delivery platforms that better enable continual systemic administration of the agonist like in our study may stimulate greater antitumor immunity and reveal more clinically efficacious outcomes.

Despite the conflicting literature surrounding the pro- or antitumorigenic role of TLR9 in prostate cancer, and consistent with the ability of CPG-ODNs to act as immunostimulatory adjuvants for vaccines [79], our study provides support for the idea that systemic stimulation of TLR9 reduces the growth of orthotopic tumors *in vivo* by boosting systemic innate immunity. This is in the context whereby the tumor cells themselves do not respond to TLR9 signaling; therefore, combining such treatment with tumor profiling to identify TLR9 nonresponsive tumors would be beneficial to avoid any potential enhancement of a tumor’s invasive properties. Alternatively, the design of TLR9 agonists that stimulate responses in specific cell types (tumor vs. immune cell) could help address this point. It is, therefore, critical that we continue to understand the biology of TLR9 as, for instance, the differential endosomal biology between cancer and noncancerous cells [80] could be exploited for this purpose. Furthermore, conducting these experiments in an orthotopic prostate tumor model allows us to test potential therapeutic strategies in a disease model that better recapitulates the highly inflammatory tumor microenvironment exhibited by prostate cancers as well as observe the effects of global immunity. Prostate cancer and its interaction with the host immune system is immensely complex, so the function of TLR9 in immune cells to negatively regulate prostate tumor pathogenesis could be considered in the development of novel prostate cancer therapeutics.

## Figures and Tables

**Figure 1 cells-13-00097-f001:**
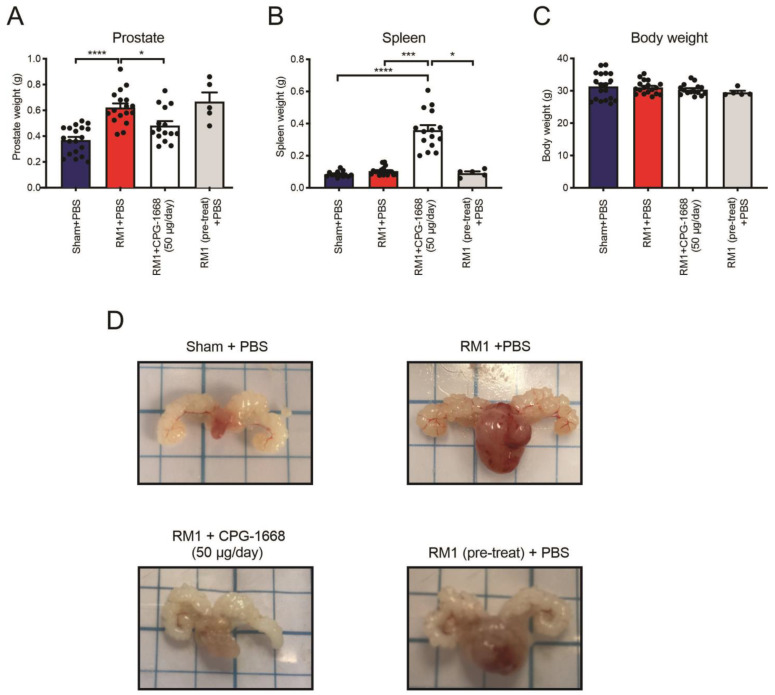
Pharmacological stimulation of TLR9 with CPG-1668 impedes murine prostate cancer tumorigenesis. C57BL6/J wild-type mice underwent engraftment surgery with a vehicle (DMEM + 10% FBS) ‘Sham’ or 5 × 10^3^ RM1 murine prostate cancer cells injected into the ventral lobe of the prostate. Some mice were injected with RM1 cells that had been pretreated with the TLR9 agonist 10 µg/mL CPG-1668 for 1 h prior to injection. Osmotic minipumps containing the vehicle (PBS) or CPG-1668 were also implanted subcutaneously on the same day. Osmotic minipumps allowed for constant infusion of the vehicle or CPG-1668 (50 µg/day) over 14 days. Mice were culled on day 14 and (**A**) prostates, (**B**) spleens, and (**C**) body weights recorded. (**D**) Representative photographs of prostates from each treatment group are shown. Data represent *n* = 15–19 per group (*n* = 5 for ‘RM1 pre-treat’ group only) and are expressed as mean ± SEM. Statistical analysis was conducted using ordinary one-way ANOVA followed by Tukey’s post hoc test for multiple comparisons (* *p* < 0.05, *** *p* < 0.001, **** *p* < 0.0001).

**Figure 2 cells-13-00097-f002:**
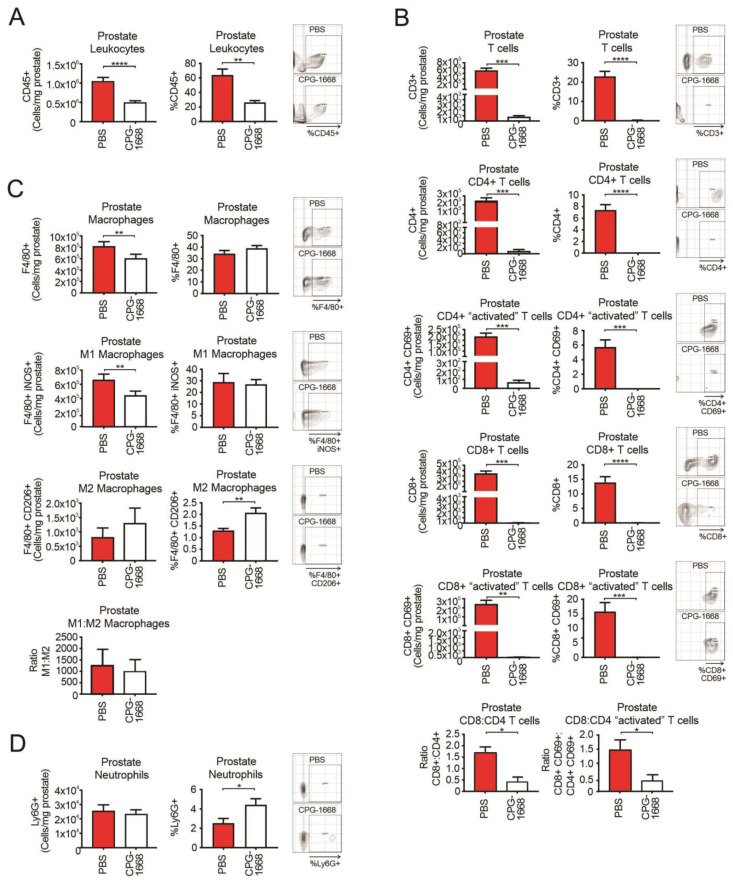
TLR9 agonism ameliorates prostate cancer-induced T-cell inflammation and reduces M1 macrophage populations within mouse prostates. Prostates from C57BL6/J mice engrafted with RM1 tumors and treated with PBS or CPG-1668 (50 µg/day) were harvested after 14 days, disaggregated, and immunophenotyped by flow cytometry. (**A**) Total CD45+ leukocytes, (**B**) CD3+, CD4+ or CD8+ T-cell populations with or without CD69 costaining, (**C**) F4/80+ total, iNOS+ (M1) or CD206+ (M2) macrophages, or (**D**) overall Ly6G+ neutrophil populations were analyzed. Data are expressed as number of cells per mg of prostate tissue or cell frequency relative to gated live cells. Representative plots for gating positive cells are shown alongside each cell type. Data represent *n* = 6 per group and are expressed as mean ± SEM. Statistical analysis was conducted using an unpaired, two-sided *t*-test (* *p* < 0.05, ** *p* < 0.01, *** *p* < 0.01, **** *p* < 0.01).

**Figure 3 cells-13-00097-f003:**
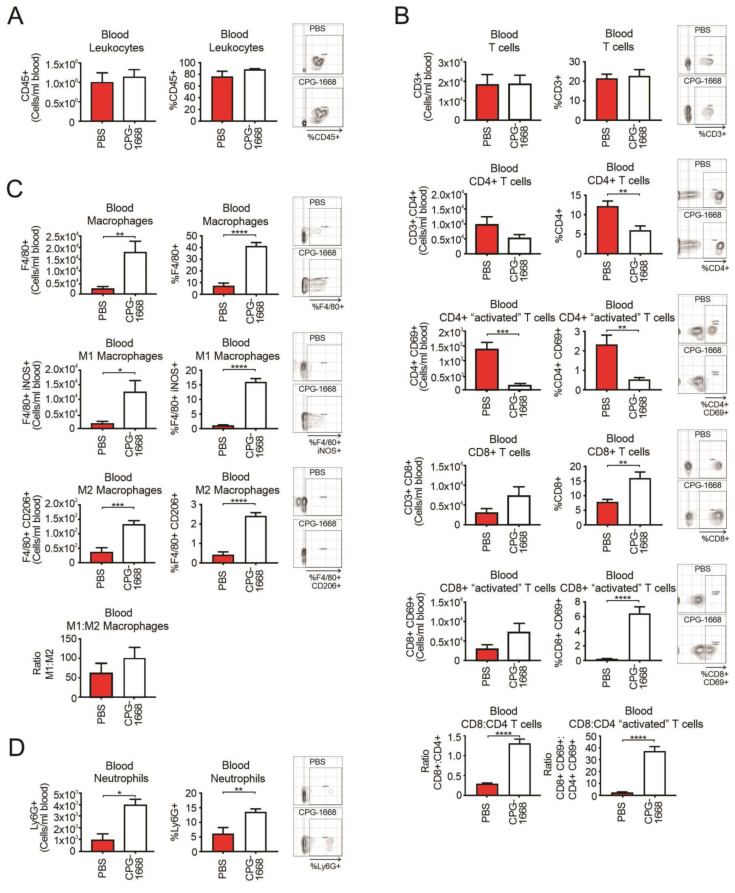
TLR9 agonism increases systemic CD8+ T-cell, macrophage, and neutrophil populations during murine prostate tumorigenesis. Blood from C57BL6/J mice engrafted with RM1 tumors and treated with PBS or CPG-1668 (50 µg/day) was collected (~1 mL) after 14 days and stained for immune-cell types for determination by flow cytometry. (**A**) Total CD45+ leukocytes, (**B**) CD3+, CD4+ or CD8+ T-cell populations with or without CD69 costaining, (**C**) F4/80+ total, iNOS+ (M1) or CD206+ (M2) macrophages, or (**D**) overall Ly6G+ neutrophil populations were analyzed. Data are expressed as number of cells per mL of blood or cell frequency relative to gated live cells. Representative plots for gating positive cells are shown alongside each cell type. Data represent *n* = 6 per group and are expressed as mean ± SEM. Statistical analysis was conducted using an unpaired, two-sided *t*-test (* *p* < 0.05, ** *p* < 0.01, *** *p* < 0.01, **** *p* < 0.01).

**Figure 4 cells-13-00097-f004:**
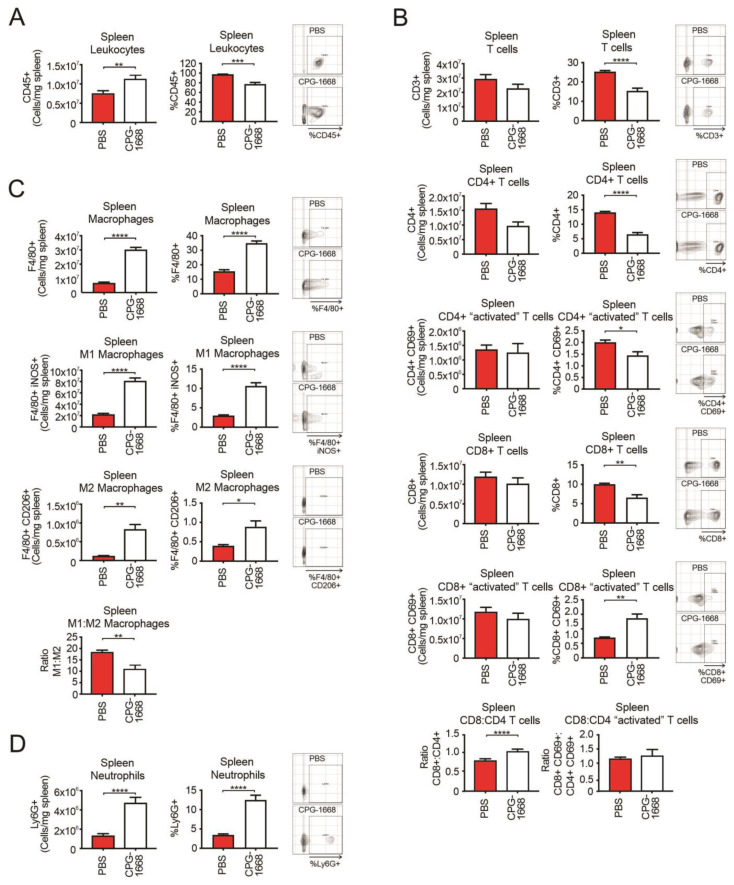
TLR9 agonism causes an influx in splenic macrophages and neutrophils but not T cells during murine prostate tumorigenesis. Spleens from C57BL6/J mice engrafted with RM1 tumors and treated with PBS or CPG-1668 (50 µg/day) were harvested after 14 days, disaggregated, and immunophenotyped by flow cytometry. (**A**) Total CD45+ leukocytes, (**B**) CD3+, CD4+ or CD8+ T-cell populations with or without CD69 costaining, (**C**) F4/80+ total, iNOS+ (M1) or CD206+ (M2) macrophages, or (**D**) overall Ly6G+ neutrophil populations were analyzed. Data are expressed as number of cells per mg of spleen tissue or cell frequency relative to gated live cells. Representative plots for gating positive cells are shown alongside each cell type. Data represent *n* = 6 per group and are expressed as mean ± SEM. Statistical analysis was conducted using an unpaired, two-sided *t*-test (** p* < 0.05, ** *p* < 0.01, *** *p* < 0.01, **** *p* < 0.01).

**Figure 5 cells-13-00097-f005:**
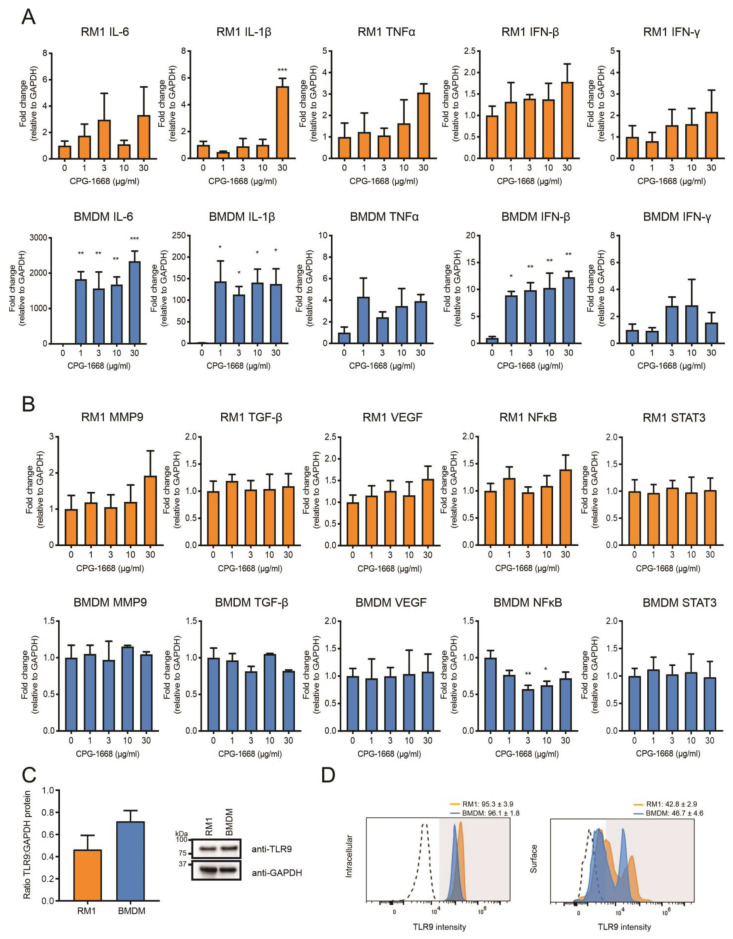
The TLR9 agonist CPG-1668 alters expression of inflammatory genes in bone-marrow-derived macrophages but not in RM1 murine prostate cancer cells. RM1 prostate cancer cells or bone-marrow-derived macrophages (BMDMs) were treated *in vitro* with increasing concentrations of CPG-1668 for 24 h. QPCR analysis of (**A**) inflammatory genes (*IL-6*, *IL-1β*, *TNFα*, *IFN-β*, and *IFN-γ*) or (**B**) tumor-promoting genes (*MMP9*, *TGF-β*, *VEGF*, *NFκB*, and *STAT3*) was performed. Data is presented relative to *GAPDH* housekeeping and expressed relative to untreated controls. (**C**) Basal expression of TLR9 or GAPDH protein in RM1 and BMDMs was determined by Western blot. Quantification of band density was performed and expressed as TLR9 relative to GAPDH. (**D**) Intracellular or surface staining of TLR9 was conducted in RM1 or BMDM cells. Positively stained cells were gated above unstained isotype controls (dotted lines). Data represent *n* = 3–4 per group and expressed as mean ± SEM. Statistical analysis was conducted using one-way ANOVA followed by Dunnett’s post hoc test for multiple comparisons (* *p* < 0.05, ** *p* < 0.01, *** *p* < 0.001).

**Figure 6 cells-13-00097-f006:**
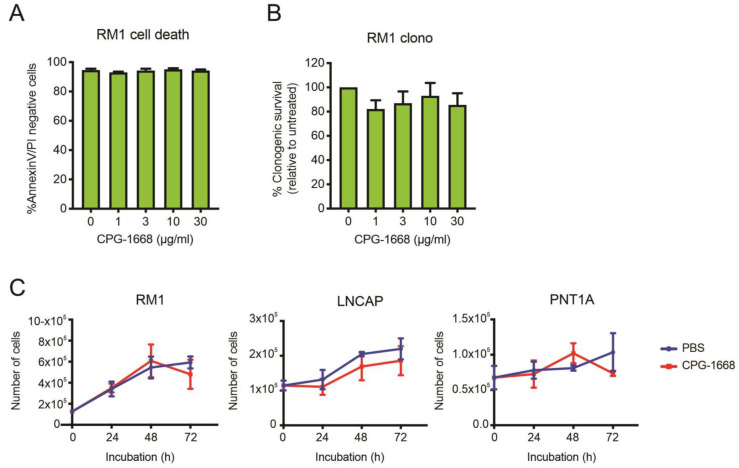
The TLR9 agonist CPG-1668 does not alter cancer cell viability. RM1 cells were treated *in vitro* with increasing concentrations of CPG-1668; then, (**A**) cell death (measured by AnnexinV-FITC and propidium iodide (PI) staining) or (**B**) colony-forming potential after 48 h or 24 h incubation, respectively, was determined. (**C**) RM1 murine prostate cancer cells, LnCAP human prostate cancer cells, or PNT1A normal human prostate cells were seeded at an appropriate density and grown in media containing the vehicle (PBS) or CPG-1668 (10 µg/mL/day) for 72 h. Cell numbers (representing total live cells) were determined every 24 h via trypan blue cell-exclusion assay and expressed as total live cells. Data represent *n* = 3–5 per group and are expressed as mean ± SEM. Statistical analysis was conducted using (**A**,**B**) ordinary one-way ANOVA, followed by Dunnett post hoc test for multiple comparisons, or (**C**) two-way ANOVA, followed by Sidak’s post hoc test for multiple comparisons (statistical significance for all conditions was *p* > 0.05).

**Figure 7 cells-13-00097-f007:**
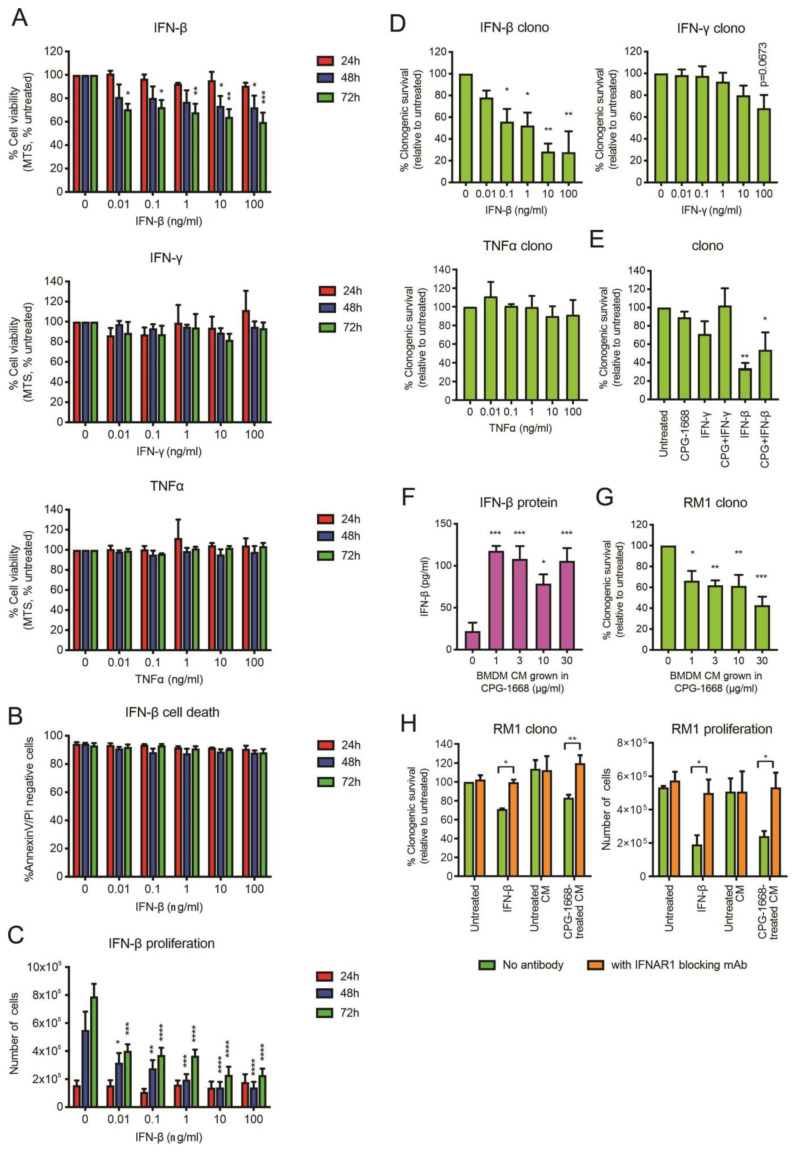
RM1 cells are sensitive to the growth inhibitory effects of IFN-β secreted by CPG-1668-stimulated macrophages. (**A**) RM1 cells were treated with various concentrations of recombinant cytokines (IFN-β, IFN-γ, or TNFα) for 24, 48, or 72 h; then, cell viability was determined by MTS absorbance. (**B**) Cell death (measured by AnnexinV-FITC and propidium iodide (PI) staining) or (**C**) proliferation (measured by the total number of live cells) was also assessed after incubation with IFN-β at the same time points. (**D**) The colony-forming potential of RM1 cells was determined after 24 h incubation with recombinant cytokines. (**E**) The clonogenicity of cells following cotreatment with CPG-1668 (10 µg/mL) and IFN-γ (100 ng/mL) or IFN-β (100 ng/mL) was also determined. (**F**) The amount of IFN-β secreted into the culture media of BMDMs grown in various concentrations of CPG-1668 was determined by ELISA. (**G**) RM1 cells were grown in BMDM-conditioned media (CM) for 24 h; then, the colony-forming potential was determined. (**H**) Some cells were pretreated with 50 µg/mL IFNAR1 blocking monoclonal antibody for 1 h before the addition of BMDM-derived CM or recombinant IFN-β (1 ng/mL). The colony-forming potential after 24 h or cell proliferation after 48 h was determined. Data represent *n* = 3–4 per group and expressed as mean ± SEM. Statistical analysis was conducted using (**D**–**G**) ordinary one-way ANOVA followed by Dunnett’s post hoc test for multiple comparisons or (**A**–**C**,**H**) two-way ANOVA followed by Sidak’s post hoc test for multiple comparisons (* *p* < 0.05, ** *p* < 0.01, *** *p* < 0.001, **** *p* < 0.0001).

## Data Availability

The data presented in this study are available in this article and the relevant Appendix A.

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
