# Peer review of "TLR9 Monotherapy in Immune-Competent Mice Suppresses Orthotopic Prostate Tumor Development"

_cells, 2024, doi:10.3390/cells13010097_

Round 1

Reviewer 1 Report (Previous Reviewer 2)

Comments and Suggestions for Authors

This new version of the manuscript is fine.

Author Response

We thank this reviewer for their assessment of our revised manuscript.

Reviewer 2 Report (Previous Reviewer 4)

Comments and Suggestions for Authors

The manuscript has improved significantly. I have only one concern left to be addressed. Many TLR9 antibodies in the market do not recognize the right protein and this has caused confusion in the field. Please validate the western blot results Fig5C with TLR9 targeting siRNA using the prostate cancer cell line to be sure that the band detected is TLR9.

Comments on the Quality of English Language

Only basic editing process needed.

Author Response

Reviewer Point: Please validate the western blot results Fig5C with TLR9 targeting siRNA using the prostate cancer cell line to be sure that the band detected is TLR9.

Author response: We thank this reviewer for raising this point. The antibody we used in this study was a monoclonal mouse anti-TLR9 antibody (clone 26C593.2) purchased from Abcam (catalogue number ab134368). This antibody has been used in various studies to measure expression of both mouse and human TLR9 in different tissues (e.g. Luo et al. Oncol Lett. 2020. doi: 10.3892/ol.2020.11971; Liu et al. BMC Gastroenterol. 2022. doi: 10.1186/s12876-022-02527-z; Zhang et al. J Neuroinflammation. 2020. doi: 10.1186/s12974-019-1673-3; Min et al. Int J Immunopathol Pharmacol. 2022. doi: 10.1177/03946320221090007; Layunta et al. J Physiol Biochem. 2022. doi: 10.1007/s13105-022-00897-2.).

Of note, the studies by Shen et al. Diabetol Metab Syndr. 2022. doi: 10.1186/s13098-021-00780-y and Zhang & Li. Exp Cell Res. 2020. doi: 10.1016/j.yexcr.2020.112159 performed siRNA-mediated downregulation of TLR9 in various mouse tissues and used this particular antibody to confirm reduced protein expression, thereby validating the specificity of the 26C593.2 clone for mouse TLR9. We therefore believe the band we observe in our Western blots represents TLR9.

This manuscript is a resubmission of an earlier submission. The following is a list of the peer review reports and author responses from that submission.

Round 1

Reviewer 1 Report

Comments and Suggestions for Authors

In the present study, Miles et al. used an orthotopic murine model of prostate cancer and examined the effect of a synthetic GpG agonist CpG-1668, primarily acting on TLR9, on prostate tumorigenesis. The authors nicely showed that the daily administration of GpG-1668 in mice inhibited the prostate tumorigenesis and development after 14 days, indicating a promising effect of TLR9 treatment for prostate cancer. However, the conclusion and the author’s statement about the underlying effects for that phenomenon are not supported by the results and should be reconsidered. 

My major concerns are: 

1.    Authors should always show gating strategies and dot blots for FACS data and not only bar graphs.

2.    Authors should use a more precise method for cell proliferation (e.g., flow cytometry).

3.    MTS is not a good indicator for cell viability but for the metabolic activity of the cells. Authors should use a more precise method for cell viability (e.g., flow cytometry).

4.     Why would CD69+ CD8 T cells promote tumor growth? Using only CD69 as marker for cell activation without any further characterization of the CD8 T cells does not allow to make a conclusion about their role in tumor progression. What is the state of the CD8 T cells? Are those effector cells, effector memory cells or central memory cells (e.g., CD44, CD62L)? Did authors check for other specific surface activation/inhibition marker on CD8 T cells such as CD25, PD1, Lag3, TIGIT, Tim3 or 2B4? Did they check for homing markers or tumor trafficking markers (e.g., CCR7, CXCR3). Did they check for their cytotoxic content (IFN, TNF, Granzyme B)?

5.    What about further characterization of the CD4+ T cells?

6.    What about tumor, systemic and splenic NK cells, B cells and especially DCs, which are known to have a high surface expression of TLR9 and are important for antigen presentation to T, B and NK cells?

7.    What about the surface TLR9 expression on T cells and macrophages in the blood, spleen and tumor?

8.    Why should the CD8 T cells in untreated tumors be restricted in their effector function? Cytotoxic CD8 T cells are important to inhibit and reduce tumor progression. I do not believe that the reduced CD8 T cell number results in reduced tumor progression. Authors should give an explanation for that phenomenon. 

9.    How could authors claim that the CD8 T cell number diminished over the time period of 14 days if they do not have other time points? I genuinely do not believe that a reduced number of T cells and macrophages in the tumor in the CpG-1668 treated animals is the reason for reduced/impaired tumorigenesis. 

10.  TLR9 unleashes the cytotoxic potential of CD8 T cells? Could authors prove that? Authors could have isolated CD8 T cells from blood, spleen, draining lymph nodes and tumor and rechallenge them with TLR9 to prove that point.  

11.  A shift in the ratio of CD8 and CD4 T cells does not indicate a change in tumor lytic response of CD8 T cells. Phenotyping of CD8 and CD4 T cells would have been necessary to support that statement. 

12.  Authors showed reduced macrophages in GpG-treated tumors but claim that macrophages release type I IFNs which then suppresses tumor growth?

13.  Why not treating the RM1 cells “long enough” with the CpG-1668 to exclude any effects of CpG-1669 on RM1 cells?

14. Have authors measured systemic cytokines/chemokines? 

Reviewer 2 Report

Comments and Suggestions for Authors

The authors provide here an interesting study about the possible use of TLR8 agonists in the therapy of prostate cancer. They identified immune response and IFNbeta as the main players.

Comments to the authors

1.       Lines 51-53 - while endosomal TLRs such as TLR3, 7, 8 and 9 activate upon the recognition of endocytosed antigens like viral nucleic acids- These endosomal TLRs can also be activated by DAMPs, please rewrite this sentence

2.       TIR is a domain of Toll-like receptors and not an adaptor protein

3.       Figure 4C. I am a bit puzzled by the ratio of M1:M2 macrophages in CPG166B treated spleen. Could you check the calculation?

4.       Line 310- I would say M2 macrophages were slightly increased but it was not statistically significant. Also, the authors do not discuss the reduction of M1 macrophages and slight increment of M2 macrophages in prostate, except in one sentence (lines 522-526) which is unclear. I suggest this sentence should be rewritten and the amount of M1 and M2 macrophages should be discussed more extensively.

5.       Figure 5A. Are the authors certain that TNFalpha (30ug/mL) is not statistically significant compared to the untreated?

6.       Line 413   was determined

7.       The authors should add representative photos of colony forming potential experiments (at least for the experiments where there is evident change).

8.       Lines 564-567 “It is possible for the RM1 cells used in our study to exhibit defective or limited TLR9 signaling although the fact that CPG-1668 also had no effect on the growth of human prostate cancer (LnCAP) and non-malignant cell lines suggests this observation was not specific to the RM1 cell line.”

It really is interesting that the authors got this since other authors previously got different results on the same cell line (for example: Ilvesaro 2007 TollLikeReceptor-9 Agonists Stimulate Prostate Cancer Invasion In Vitro- LnCap- Increased invasion)

Are the authors certain that they have the proper cell lines? Has the authentication been performed?

9.       Please check the reference style recommended by the Journal (DOI?)

Reviewer 3 Report

Comments and Suggestions for Authors

The manuscript explores the potential of TLR9 as a monotherapy for prostate cancer in a murine model. The overall communication is effective, but it would be beneficial to clarify the clinical implications of the findings. Additionally, providing more context on the conflicting literature regarding TLR9 in prostate cancer could enhance understanding. The statement about tumor cells not responding to TLR9 signaling is intriguing; however, a brief discussion on the implications for potential therapies would add value.

In the discussion section, a deeper insight into how the observed responses of CD8+ T cells and M1 macrophages might influence or shape potential treatments for prostate cancer would be helpful.

Reviewer 4 Report

Comments and Suggestions for Authors

Miles et al. have studied the effects of TLR9-activating dinucleotides in prostate cancer syngeneic murine model. They suggest that synthetic TLR9 agonist CPG-ODN (CPG-1668) provokes a TLR9 mediated immune response and impairs prostate tumorigenesis. Novelty here is that they used an orthotopic syngeneic model which represents better the natural microenvironment than previously published subcutaneous models or in vitro systems.

This is a well-written manuscript with considerable amount of experimentation also in vivo, and an important topic, but there is one big major concern: TLR9 is activated here by unmethylated cytosine-guanine dinucleotide (CPG) sequences, and it is fundamental to use control other than sham for CPG-1668 that does not activate TLR9 at least in some of the key experiments. The comparison here is only untreated tumor, but some experimental data should absolutely be provided that contains this control, a similar dinucleotide that does now activate TLR9 or by other means prove that TLR9 is necessary for the observed phenomena. This data is important because previous publications have been contradictory, and carefully controlled experiments will help the field to make conclusions regarding the potential of TLR9 agonism in cancer. Now it is difficult to conclude that the effects are mediated via TLR9.

Add the catalogue and lot numbers for antibodies used (e.g. lanes 236-237).

Comments on the Quality of English Language

The manuscript is well-written.

Reviewer 5 Report

Comments and Suggestions for Authors

Employing a syngeneic murine model of prostate cancer, authors demonstrate that by using a synthetic agonist of TLR9, CPG-1668, TLR9 mediated immune response is evoked that leads to impairment of prostate tumorigenesis. They demonstrate that untreated tumors lack anti-tumor activity while pharmacological activation of TLR9 enhanced the abundance of CD8+ T cells and M1 macrophages, leading to inhibition of tumor growth. The role of TLR9 signaling in prostate cancer is not fully known. Therefore, the present study aims to address this knowledge gap using a syngeneic mouse model of prostate cancer.  However, the study employs a single model for prostate cancer. They performed in vitro experiments to show the lack of effects on proliferation in prostate cancer cell lines. It would be more convincing to take tumors from in vivo studies represented in Fig. 1 and stain for proliferative markers. Previous studies in the field have shown an opposite effect of TLR9 on prostate cancer via effects on STAT signaling  ( PMC5505743).  Authors should check the expression of STAT3 and LIF in their model. 

Minor: 

Actual p-vlaues for all figures should be reported. 
